# Research on emergency material demand based on urgency and satisfaction under public health emergencies

**Zhichao Ma[¤], Jie Zhang[¤]\*, Shaochan Gao[¤]**

School of Management Engineering and Business, Hebei University of Engineering, Handan, Hebei, China

¤ Current address: Hebei University of Engineering, Congtai District, Handan, Hebei, China
\* zhangjie716@126.com

**Data Availability Statement:** All relevant data are within the paper and its Supporting information files.

**Funding:** This research was funded by the National Natural Science Foundation of China (Grant No.

## Abstract

In recent years, the frequent occurrence of public health emergencies has had a significant impact on people's life. The study of emergency logistics has also attracted scholars' attention. Therefore, matching emergency materials' supply and demand quickly, which meets urgency and satisfaction, is the purpose of this paper. This paper used the Metabolism Grey Model (1,1) (GM (1,1)) and the material demand prediction model to predict the number of infections and material demand. Besides, we established a bi-objective optimization model by constructing a profit and loss matrix and a comprehensive utility perception matrix. The results show that the method is helpful in matching the optimal supply and demand decision quickly on the basis of meeting urgency and satisfaction. The method is helpful in improving the fairness of emergency material distribution, which could better protect people's livelihoods.

## 1. Introduction

Public health emergencies refer to the public health risks, caused by the spread of diseases to a country or region. The rapid spread poses a great threat to people's health. Therefore, rapid prediction and decision-making of public health emergencies are particularly important. Emergency logistics is an important guarantee for public health emergencies. Therefore, the research on public health emergencies and emergency logistics has attracted more scholars.

At present, the research on emergency logistics mainly focuses on the following aspects: emergency materials' path optimization, emergency materials' demand forecasting and emergency materials' distribution. Identifying whether emergency materials are accurately and efficiently to the destination is a research hotspot. Based on this, the programming model of emergency logistics support system was proposed by Özdamar et al. [1]. The study shows that the model can predict the future supply time of goods and provide a solution for repeated transportation of vehicles. Chang et al. proposed a multi-objective genetic algorithm. The research shows that the model can generate a fast and efficient rescue material scheme with the consideration of transportation speed and cost [2–4]. In order to better manage the risk of disaster situations, Li et al. proposed a two-stage model [5]. The study shows that using Bayesian network and Monte Carlo method to locate emergency stations can help speed recovery

71871084) and Social Science Grand Research of the Hebei Education Department (Grant No. BJ2021070). The funders had no role in study design, data collection and analysis, decision to publish, or preparation of the manuscript.

**Competing interests:** The authors have declared that no competing interests exist.

from disasters [6]. Liu et al. used dynamic programming algorithm to provide decision-making basis for material allocation and supply sequence allocation in the background of medical materials scheduling in public health emergencies. The study will help optimize the distribution of emergency supplies and make planning more realistic [7, 8]. Liu et al. build a multi-objective mathematical model of emergency materials distribution. The research provides a solution for the optimization of emergency vehicle distribution path [9–11].

The prediction of emergency logistics is the premise and foundation of the optimal allocation of emergency materials. Sun et al. put forward two universal fuzzy-rough set models. The model aims to reduce the influence of uncertain factors on decision-making environment and improve the accuracy of prediction [12]. To improve the accuracy and stability of prediction, Chen et al. proposed a new, improved algorithm based on the three-stage ant colony optimization- Back Propagation (IACO-BP) algorithm. The algorithm accurately and efficiently predicts emergency material demand and can better assist the disaster situation [13]. Zhu et al. believed intelligent information processing technology can help adapt to dynamic demand forecasting scenarios in emergency and rescue situations. The research shows that intelligent information processing technology is helpful to real-time prediction in the emergency scene [14]. By combining the Dempster-Shafer and cosmic background radiation (CBR) methods, Fei et al. proposed a natural disaster scenario matching method and constructed a dynamic forecasting model of demand for emergency materials. The results show that the prediction results of natural disaster losses help improve the efficiency of emergency disaster response [15]. Shao et al. put forward a method of demand forecasting based on intuitionistic fuzzy case-based reasoning (IFCBR), aiming at the characteristics of incomplete, inaccurate, and uncertain demand forecasting information for disaster relief supply. The research shows that this forecasting method provides decision support for the demand for relief materials and the basis for resource allocation [16, 17].

There are also many methods to study the distribution of emergency supplies. For example, Yang et al. proposed a robust optimization model for the multi-stage dynamic pre-positioning allocation of emergency materials. The study shows that separating the static pre-disaster stage and the dynamic post-disaster stage helps improve the scientific distribution of emergency materials [18]. Based on the robust parameter optimization method, Bai et al. divided the emergency materials allocation problem into six mixed integer parameter submodels. The study shows that the parameter-based domain decomposition method can make the actual problem more mathematical and improve decision-making efficiency [19]. In order to realize the efficient distribution of three-dimensional (3D)-printed emergency materials, He et al. proposed an improved non-inferiority sorting genetic algorithm (NSGA-II). The research shows that the advantages of rapid manufacturing and a low 3D-printing threshold help alleviate the over-supply of emergency materials [20, 21]. To improve the stability of the emergency logistics system, Ke et al. used a two-stage robust optimization method to determine the location of emergency facilities and manage the emergency response system [22].

The existing literature mainly focuses on the transportation route and location of emergency logistics. However, there are few studies on the demand forecasting of emergency materials and matching supply and demand based on people's needs and satisfaction. In the context of public health emergencies, the research on emergency material mobilization and vehicle distribution route selection is relatively mature. However, in material distribution, it is easy to cause uneven material distribution due to various factors. Based on this, this paper focuses on the urgency of material demand and the satisfaction of supply and demand matching to alleviate the uneven distribution of materials.

This paper has two innovations. Firstly, the Metabolic GM (1,1) model is used to dynamically predict the number of infections, and the material demand is predicted according to the

number of infections. Secondly, based on comprehensively considering the urgency of material demand and the satisfaction of both supply and demand, the profit and loss matrix and the perceived utility matrix are constructed. Moreover, the bi-objective optimization model is used to make the optimal decision on the supply and demand matching problem of emergency materials.

The rest of the organization of this article is as follows. Section 2, materials and methods, describes the metabolic GM (1,1) model, the material demand forecasting formula and the bi-objective optimization model. Section 3, results, gives an example, calculates the forecast, and obtains the result of the optimal supply-demand matching decision. Section 4, discuss and describe the impact of different indicator weights on supply and demand sides. Section 5, conclusion, summarizes the main research results, significance, innovation and future research direction.

## 2. Materials and methods

### 2.1. Related theory introduction

In recent years, the occurrence of public health emergencies has had a significant impact on people's production and life. Because the outbreak of public health emergencies is uncertain and hidden, and the data has no clear rules, we use the Metabolism GM (1,1) model to predict the number of infections in public health emergencies. The GM (1,1) model is the basic model of the Grey System theory. Its essence is to build differential, difference, and approximate exponential law-compatible equations after generating the original sequence cumulatively [23]. However, GM (1,1) relies too much on the original data information and needs to fully consider the influence of new information on the prediction results in the medium and long-term prediction [24]. The metabolic GM (1,1) model replaces the old with the new based on metabolism [25]. This paper uses the Metabolism GM (1,1) model to predict the number of infected persons. Then according to the number of infected people, the formula of material requirement is used to calculate the amount of material requirement.

Disappointment theory is a behavioral decision theory, which is a psychological perception caused by comparing the actual value with the expected value. The disappointment theory shows that the decision maker will be ecstatic when the actual result is higher than the expected return. The decision maker will be disappointed when the actual result is lower than the expected return. The more significant the gap between the actual result and the expected return, the greater the disappointment [26, 27]. At present, there are three main disappointment models: the 'modified expected utility' disappointment theory model, the disappointment aversion model, and the unexpected disappointment theory model [28–30]. This paper uses the perceived utility matrix quantified by the disappointment-happiness function in the unexpected disappointment theory model. This method can quantify the satisfaction value of both supply and demand and promote the maximization of satisfaction.

This paper also applies the bilateral matching theory. The problem originated from the marriage matching problem, which means that two participants who do not intersect in the market match each other to meet the requirements of both parties as much as possible. A matching method that finally completes market trading activities [31–33]. This paper transforms the allocation problem after public health emergencies into a supply and demand matching problem that maximizes the utility of both supply and demand. In this study, they are used to represent the supply and demand sides of emergency material allocation. The set of supply subjects is the set of demand subjects. Regarding emergency material supply and demand matching, the multi-attribute information of emergency material demand is essential in supply and demand matching. For example, suppliers consider location, delivery capacity,

delivery time, and delivery cost as evaluation indicators. As evaluation indicators, the demand side needs to consider community conditions, family conditions, material needs urgency, transportation convenience, and material distribution. Then, comparing the evaluation expectations of suppliers and demanders, the satisfaction information is quantified to obtain the matching result of maximizing satisfaction.

## 2.2. Model building

**2.2.1. The Metabolism GM (1, 1) model.** This paper uses the metabolic GM (1,1) model to predict the number of infections under public health emergencies [34]. Taking COVID-19 as an example, this paper discusses the decision-making problem of optimal supply and demand matching. We take five supply points and five community demand points in Handan by a case study based on considering the urgency of emergency material demand and the satisfaction of both supply and demand sides. Among them is the number of infected data from the Hebei Provincial Health Commission (2022.04.05–04.11, http://wsjkw.hebei.gov.cn/).

The nonnegative original time sequence $Q^{(0)}$ and 1-AGO time series $Q^{(1)}$ showed as follows:

$$Q^{(0)} = (q^{(0)}(1), q^{(0)}(2), \ldots, q^{(0)}(t)) \tag{1}$$

$$Q^{(1)} = (q^{(1)}(1), q^{(1)}(2), \ldots, q^{(1)}(t)) \tag{2}$$

The letter $t$ represents the time. Moreover, we can see the following.

$$q^{(1)}(k) = \sum_{i=1}^{k} q^{(0)}(i), k = 1, 2, \ldots, t \tag{3}$$

$$Z^{(1)} = (z^{(1)}(2), z^{(1)}(3), \ldots, z^{(1)}(t)) \tag{4}$$

The formula (4) is the adjacent mean generating its sequence $(Q^{(1)})$.

$$z^{(1)}(k) = \frac{1}{2}\left(q^{(1)}(k) + q^{(1)}(k-1)\right), k = 2, \ldots, t \tag{5}$$

$$q^{(0)}(k) + az^{(1)}(k) = b \tag{6}$$

Eqs (3), (5), and (6) are the process of the traditional GM (1,1) model. Set the Eq (9) to a parameter column.

$$Y = \begin{bmatrix} q^{(0)}(2) \\ q^{(0)}(3) \\ \vdots \\ q^{(0)}(t) \end{bmatrix} \tag{7}$$

$$B = \begin{bmatrix} -z^{(1)}(2) & 1 \\ -z^{(1)}(3) & 1 \\ \vdots & \vdots \\ -z^{(1)}(t) & 1 \end{bmatrix} \tag{8}$$

The least square estimates of $a$ and $b$ from Eq (6) are given by

$$\hat{a} = (a, b)^T = (B^T B)^{-1} B^T Y \tag{9}$$

Using these estimates of $a$ and $b$, now our whitenization equations of the grey system (6) are given by

$$\frac{dq^{(1)}}{dt} + aq^{(1)} = b \tag{10}$$

The time response sequences and reduction values are Eqs (11), (12) and (13).

$$q^{(1)}(t) = \left( q^{(1)}(1) - \frac{b}{a} \right) e^{-at} + \frac{b}{a} \tag{11}$$

$$\hat{q}^{(1)}(k+1) = \left( q^{(0)}(1) - \frac{b}{a} \right) e^{-ak} + \frac{b}{a} \tag{12}$$

$$\hat{q}^{(0)}(k+1) = \alpha^{(1)} \hat{q}^{(1)}(k+1) \tag{13}$$

$$= (1 - e^a) \left( q^{(0)}(1) - \frac{b}{a} \right) e^{-ak}$$

In these formulas, they satisfy the condition k = 1, 2, . . ., $t$. The original sequence is in equal dimensions. Then we can remove the old information $q^{(0)}(1)$ [35]. The latest information $q^{(0)}(t+1)$ is supplemented to obtain a new sequence $Q^{(0)} = (q^{(0)}(2), q^{(0)}(3), . . ., q^{(0)}(t+1))$, thus forming a new metabolic GM (1,1) prediction model (14).

$$q^{(0)}(k) + az^{(1)}(k) = b \tag{14}$$

We deal with the second sequence in the same dimension, which removes the old information $q^{(0)}(2)$, and adds the latest information $q^{(0)}(t+2)$. Then we could obtain a new sequence:

$$Q^{(0)} = (q^{(0)}(3), q^{(0)}(4), . . ., q^{(0)}(t+2)) \tag{15}$$

We can use this as a new original sequence to calculate through continuously eliminating the old information, and adding the new information. Then we could get a new Metabolism GM (1,1) model.

The residual sequence is the Eq (16).

$$\begin{aligned}
\varepsilon^{(0)} &= (\varepsilon^{(0)}(1), \varepsilon^{(0)}(2), . . ., \varepsilon^{(0)}(t)) \\
&= ((\hat{q}^{(0)}(1), \hat{q}^{(0)}(2), . . ., \hat{q}^{(0)}(t)) - (q^{(0)}(1), q^{(0)}(2), . . ., q^{(0)}(t))
\end{aligned} \tag{16}$$

The model calculated the mean absolute percentage error (MAPE) (17).

$$\text{MAPE} = 100\% \frac{1}{n} \sum_{k=1}^{n} \left| \frac{q^{(0)}(t) - \hat{q}^{(0)}(t)}{q^{(0)}(t)} \right| \tag{17}$$

The smaller MAPE value has a better result. The accuracy was better when the MAPE value was less than 10%. If the MAPE is 0%, it indicates a perfect model. If the MAPE is more significant than 100%, it indicates the opposite.

**2.2.2. The material demand forecasting model.** In this part, we use the forecast formula of emergency materials, according to the forecast result of the number of infected people, to

**Table 1. Main parameter settings.**

| Parameter | $a_i$ (g/person.day) | $\bar{T}$ (Day) | $\alpha$ | $Z_{1-\alpha}$ |
|---|---|---|---|---|
| Numerical value | 300–500 | 1 | 0.05 | 1.65 |

forecast the number of materials needed. And the formula (18) required the specific indicators are shown in Table 1.

As shown in Table 1,The cumulative number of infected people is the $Q(t)$, which is in a large-scale epidemic (the data came from the demographic data of Handan). When a large-scale outbreak broke out in the region, the demand for supplies increased rapidly. However, it takes a certain amount of time to deploy materials [36, 37]. Therefore, the emergency material models to forecast the emergency material are formulas (18), (19) and (20).

$$D_i(t) = a_i \times Y_i(t) \times \bar{T} + z_{1-\alpha} \times STD_i(t) \times \sqrt{\bar{T}} \tag{18}$$

$$= a_i \times Q(t) \times \bar{T} + z_{1-\alpha} \times STD_i(t) \times \sqrt{\bar{T}}$$

$$STD_i(t) = \sqrt{\frac{\sum_{k=0}^{t-1} [D_i(t-k) - \bar{D}_i(t)]^2}{t-1}} \tag{19}$$

$$\bar{D}(t) = \frac{a_i \times \sum_{k=0}^{t-1} Q_i(t-k)}{t} \tag{20}$$

$i$ refers to emergency supplies, $a_i$ represents the demand for goods $i$ per person in unit time. $D_i(t)$ is the demand for the type of emergency supplies $i$ in the epidemic area. $\bar{T}$ is the upper bound of the time interval between the two shipments. $Z_{1-\alpha}$ represents the safety factor if people in the epidemic area tolerate a goods shortage $\alpha$. $STD_i(t)$ is the instantaneous change of emergency supplies in the time $\bar{T}$. $\bar{D}_i(t)$ is the average value of the time-varying demand for emergency supplies within the predicted time $t$.

**2.2.3. Decision making model.** *(1). Establish the supply and demand indicators system.* Based on the forecast of material demand, this paper further studies the urgency of material demand. In this paper, according to the urgency of material demand factors, we established the evaluation index system of supply and demand sides. The details are shown in Tables 2 and 3.

Both Tables 2 and 3 established the indicator system based on the impact factors of demand urgency. Table 2 was from the demand side perspective, including community, family, material needs emergency, traffic situation, and distribution of five aspects. One of the prerequisites of this study is to be in the same region. Supply capacity, emergency materials reserve, and distribution costs belong to the cost-type indicators. That is, the greater the impact factor, the lower the score, and the lower the urgency of material needs.

Table 3 shows the attributes of the supply-side indicators, which focus on two types of indicators: community and distribution. In general, the closer the location to the community, the shorter the distribution time, the lower the cost of distribution, and the lower the urgency of material needs. And the weaker the supply capacity, the higher the urgency of material demand.

**Table 2. The demand side establishes the indicator attribute.**

| First-class index | Second-class index | Original index value | Index type |
|---|---|---|---|
| Community | Location | Distance from the point of supply less than 500 meters is 1; 500–1,000 meters is 2; 100–1,500 meters is 3; 1500–2,000 meters is 4; more than 2,000 meters is 5. | Efficiency-type indicators |
|  | Supply ability | The supply capacity is fragile for 5; it is weak for 4; it is generally 3; it is vital for 2; it is strong for 1. | Cost-based indicators |
| Family | Difficult number (old, weak, disabled, pregnant) | A number greater than or equal to five is 5; four-five is 4; three-four is 3; two-three is 2; a number less than or equal to one is 1. | Efficiency-type indicators |
|  | Population | When the number is more than five, we assign it 5; four-five is 4; three-four is 3; two-three is 2; one-two is 1. | Efficiency-type indicators |
|  | The number of affected people | The number of affected persons greater than five is 5; five-four is 4; four-three is 3; three-two is 2; two-one is 1. | Efficiency-type indicators |
| Emergency situation of material requirement | Status of emergency supplies | Days for reserve material used are 1 for more than fifteen days; 2 for fifteen-ten days; 3 for ten-five days; 4 for five-one days; 5 for less than one day. | Cost-based indicators |
|  | Non-replaceable items out of stock time | When the length is less than one day is 1; one-three days is 2; three-five days is 3; five-seven days is 4; more than seven days is 5. | Efficiency-type indicators |
| Traffic conditions | Traffic convenience | The distance between the location and the traffic hub, which is less than 500 meters, is 1; 500–1000 meters is 2; 1000–1500 meters is 3; 1500–2000 meters is 4; more than 2000 meters is 5. | Efficiency-type indicators |
| Distribution situation | Delivery cost | Distribution costs are highest at 1, higher at 2, moderate at 3, lower at 4, lowest at 5. | Efficiency-type indicators |
|  | Delivery time | The distribution time is 5 for shortest, 4 for shorter, 3 for moderate, 2 for longer, 1 for longest. | Efficiency-type indicators |

*(2). The index weight is calculated by entropy weight method.* The entropy weight method effectively calculates the weight of multi-attribute indexes and evaluates the indexes according to the entropy value of data information. In this study, the entropy weight method is used to calculate the index weight of the urgency degree of the emergency materials demand of both the supplier and the demander [38].

**Table 3. The provider metric attribute.**

| First-class index | Second-class index | Original index value | Index type |
|---|---|---|---|
| Community | Location | Distance from the point of supply less than 500 meters is 1;<br>500–1,000 meters is 2;<br>100–1,500 meters is 3;<br>1500–2,000 meters is 4;<br>more than 2,000 meters is 5. | Efficiency-type indicators |
| | Supply ability | The supply capacity is fragile for 5;<br>it is weak for 4;<br>it is generally 3;<br>it is vital for 2;<br>it is strong for 1. | Cost-based indicators |
| Distribution situation | Delivery cost | Distribution costs are highest at 1,<br>higher at 2,<br>moderate at 3,<br>lower at 4,<br>lowest at 5. | Efficiency-type indicators |
| | Delivery time | The distribution time is 5 for shortest,<br>4 for shorter,<br>3 for moderate,<br>2 for longer,<br>1 for longest. | Efficiency-type indicators |

1. Calculating the standardized judgment matrix $Y$:

$$Y_{ij} = \frac{x_{ij} - \min(x_{ij})}{\max(x_{ij}) - \min(x_{ij})} \tag{21}$$

$x_{ij}$ is the index original value and $Y_{ij}$ is the standardized value.

2. The entropy $E_j$ of each index is calculated respectively, and the formula is as follows:

$$p_{ij} = \frac{Y_{ij}}{\sum_{i=1}^{n} Y_{ij}} \tag{22}$$

$$E_j = -\ln(n)^{-1} \sum_{i=1}^{n} p_{ij} \ln p_{ij} \tag{23}$$

In the formula, $p_{ij}$ is the proportion of the $j$ evaluation factor in the $j$ evaluation index, $E_j$ is the index information entropy.

3. Calculating index weight $\alpha_j$:

$$\alpha_j = \frac{1 - E_j}{\sum_{j=1}^{k} (1 - E_j)} \tag{24}$$

$\sum_{j=1}^{k} \alpha_j = 1, 0 \leq \alpha_j \leq 1$ is satisfied in the formula.

*(3). Build a profit and loss matrix*. In this study, the profit and loss value of the supplier and demander can be described as follows. When the expectation value of the demander $T_i$ for the

attribute $C_l$ of emergency supplies is inferior to that of the supplier $L_j$, it is a gain; otherwise, a loss [39]. The above gains and losses are quantified below.

Let $E_j = [d_{ijl}]_{m \times n}$ be the profit and loss matrix of the demand subject $T_i$ under attribute $C_l$. Moreover, $d_{ijl}$ is the profit and loss value of the demand subject $T_i$ and supply subject $L_j$. There are the following determination methods. When attribute $C_l$ is the distribution cost, in order to ensure the fairness of emergency materials distribution, it is set as the middle value of the expected demand subject $T_i$ and the expected supply subject $L_j$. Then the formula of distribution cost $p_{ij}$ is:

$$p_{ij} = \frac{r_{jl} + e_{il}}{2}, i \in M, j \in N, l \in K \tag{25}$$

When attribute $C_l$ is the distribution cost, $d_{ijl}$ can be divided into three cases:

Case 1: when $e_{il} = r_{jl}$, the performance of demand subject $T_i$ to supply subject $L_j$ has neither gain nor loss, $d_{ijl} = 0$.

Case 2: When $e_{il} < p_{ij} < r_{jl}$, the demand subject $T_i$ has a loss to the supply subject $L_j$, $d_{ijl} = 1 - \frac{p_{ij}}{e_{il}}$.

Case 3: when $e_{il} > r_{jl}$, the demand subject $T_i$ is the income to the supply subject $L_j$, $d_{ijl} = 1$.

Thus, when the attribute $C_l$ is the distribution cost, the formula for calculating $d_{ijl}$ can be expressed as:

$$d_{ijl} = \begin{cases} 0, e_{il} = r_{jl} \\ 1 - \dfrac{p_{ij}}{e_{il}}, e_{il} < p_{ij} < r_{jl}, i \in M, j \in N, l \in K \\ 1, e_{il} > p_{ij} > r_{jl} \end{cases} \tag{26}$$

When the attribute $C_l$ is not a distribution cost, consider the following two forms of constraint.

When the requirement body $T_i$ is a benefit constraint with respect to attribute $C_l$, the formula for calculating $d_{ijl}$ is as follows:

$$d_{ijl} = \begin{cases} 0, r_{jl} = e_{il} \\ r_{jl} - e_{il}, r_{jl} \neq e_{il} \end{cases}, i \in M, j \in N, l \in K \tag{27}$$

When the requirement body $T_i$ is a cost constraint concerning attribute $C_l$, the cost constraint means that the smaller the expected level of the attribute, the better, $d_{ijl}$ is calculated as follows:

$$d_{ijl} = \begin{cases} 0, r_{jl} = e_{il} \\ e_{il} - r_{jl}, r_{jl} \neq e_{il} \end{cases}, i \in M, j \in N, l \in K \tag{28}$$

Similarly, for the supplier, it is necessary to consider whether the expected distribution cost is within the acceptable range. Location, distribution cost and distribution time are benefit constraints, and supply capacity is a cost constraint. Let $F_l = [a_{ijl}]_{m \times n}$ be the profit and loss matrix of supply subject, where $a_{ijl}$ is the profit and loss value of supply subject $L_j$ concerning

the demand subject $T_i$. When $C_l$ is the cost of distribution, $a_{ijl}$ is calculated as follows.

$$a_{ijl} = \begin{cases} 0, e_{il} = r_{jl} \\ \dfrac{p_{ij}}{r_{jl}} - 1, e_{il} < p_{ij} < r_{jl}, i \in M, j \in N, l \in K \\ 1, e_{il} > p_{ij} > r_{jl} \end{cases} \tag{29}$$

When attribute $C_l$ is a benefit-type indicator, $a_{ijl}$ is calculated as:

$$a_{ijl} = \begin{cases} 0, r_{jl} = e_{il} \\ e_{jl} - r_{il}, r_{jl} \neq e_{il} \end{cases}, i \in M, j \in N, l \in K \tag{30}$$

When attribute $C_l$ is a cost indicator, $a_{ijl}$ is calculated as:

$$a_{ijl} = \begin{cases} 0, r_{jl} = e_{il} \\ r_{jl} - e_{il}, r_{jl} \neq e_{il} \end{cases}, i \in M, j \in N, l \in K \tag{31}$$

Because of the different dimensions of emergency materials, the profit and loss matrix $E_l = [a_{ijl}]_{m \times n}$ and $F_l = [a_{ijl}]_{m \times n}$ are transformed into standardized matrix $E'_l = [d'_{ijl}]_{m \times n}$ and $F'_l = [a'_{ijl}]_{m \times n}$, the formulas for $d'_{ijl}$ and $a'_{ijl}$ are:

$$d_{ijl}' = \frac{d_{ijl}}{\max|d_{ijl}|}, i \in M, j \in N, l \in K \tag{32}$$

$$a_{ijl}' = \frac{a_{ijl}}{\max|a_{ijl}|}, i \in M, j \in N, l \in K \tag{33}$$

*(4) Construct the perceived utility matrix.* Perceived value is the evaluation value of a product by comparing its expected value with the actual value. For the supply and demand subjects, there will be gains and losses in matching the distribution order, which is determined by psychological perception. The psychological perception was closely related to the satisfaction degree of both subjects in the final matching scheme [40]. Given this, the study constructs the expression $\varphi(x)$ of disappointment and delight according to the disappointment theory. It transforms the standardized profit and loss matrix $E'_l = [d'_{ijl}]_{m \times n}$ and $F'_l = [a'_{ijl}]_{m \times n}$ of the supply and demand sides into the perceived utility matrix $V^b_l = [v^b_{ijl}]_{m \times n}$ and $V^s_l = [v^s_{ijl}]_{m \times n}$. This may be expressed as follows.

$$\varphi(x) = \begin{cases} 1 - \alpha^x, x \geq 0 \\ \beta^{(-x)} - 1, x < 0 \end{cases} \tag{34}$$

When $\alpha$ is disappointed-happy parameter, which satisfies $0 < \alpha < 1$. And $\beta$ is disappointed-elusion parameter, which satisfies $0 < \beta < 1$. For the sake of calculation, Laciana measured $0.7 \leq \alpha \leq 0.9$ and $0.7 \leq \beta \leq 0.9$ by the behavior of most subjects [41]. In this study, for the sake of analysis, $\alpha$ and $\beta$ are usually taken as 0.8, so the formulas for calculating

$V_l^b = [v_{ijl}^b]_{m \times n}$ and $V_l^s = [v_{ijl}^s]_{m \times n}$ are:

$$v_{ijl}^b = \varphi(d_{ijl}'), i \in M, j \in N, l \in K \tag{35}$$

$$v_{ijl}^s = \varphi(a_{ijl}'), i \in M, j \in N, l \in K \tag{36}$$

According to the perceived utility matrix $V = [v_{ijl}^b]_{m \times n}$ of the demand subject $T_i$, the formula for $v_{ij}^b$ is

$$v_{ij}^b = \sum_{l=1}^{k} w_{il} v_{ijl}^b, i \in M, j \in N, l \in K \tag{37}$$

Similarly, we construct the comprehensive perceived utility matrix $V' = [v_{ij}^s]_{m \times n}$ of the supply subject $L_j$, and calculate the formula (38) as follows:

$$v_{ij}^s = \sum_{l=1}^{k} w_{jl} v_{ijl}^s, i \in M, j \in N, l \in K \tag{38}$$

In constructing the comprehensive perceived utility matrix, when $v_{ij}^b$ and $v_{ij}^s$ are bigger, the satisfaction of both the supplier and the demander is higher.

*(5) Establish the bi-objective optimization model.* Let $x_{ij}$ denote a 0–1 variable, where $x_{ij} = 0$ denotes demand that subject $T_i$ and supply subject $L_j$ does not match. Represents $x_{ij} = 1$ matching of demand subject $T_i$ and supply subject $L_j$ [42]. According to the comprehensive perceived utility matrix $V$ and $V'$, the bi-objective optimization model (39) and (40) can be established. Which is to match the supply and demand utility maximization problem of materials.

$$\max Z_1 = \sum_{i=1}^{m} \sum_{j=1}^{n} v_{ij}^b x_{ij}, i \in M, j \in N \tag{39}$$

$$\max Z_2 = \sum_{i=1}^{m} \sum_{j=1}^{n} v_{ij}^s x_{ij}, i \in M, j \in N \tag{40}$$

s.t.

$$\sum_{i=1}^{m} x_{ij} \leq 1, j \in N \tag{41}$$

$$\sum_{j=1}^{n} x_{ij} = 1, i \in M \tag{42}$$

$$x_{ij} = 0 \, or \, 1, i \in M, j \in N \tag{43}$$

Among them, Eqs (39) and (40) are the objective functions, which means that the supply-side and demand-side comprehensive perceived utility values are maximized as far as possible. Eqs (41) and (42) are the constraints of bilateral matching. Because $m \leq n$, Eq (41) is an inequality constraint. Each demand subject $T_i$ can match at most one supply subject $L_j$, and Eq (42) is an equality constraint. The implication is that each supply subject $L_j$ must and can only match one demand subject $T_i$. Since the five communities in this paper are in the same region, there are no constraints on the region.

In order to solve the multi-objective optimization model and maximize the overall matching degree, the linear weighting method is used to weigh Eqs (39) and (40). Where $w_1$ and $w_2$ represent the weights of $Z_1$ and $Z_2$ respectively, if $0 \leq w_1, w_2 \leq 1$ and $w_1 + w_2 = 1$ are satisfied. In this study, we set $w_1 = w_2 = 0.5$. And then the two-objective model's Eqs (39) and (40) can be transformed into the single-objective optimization Model (44).

$$\max Z = 0.5Z_1 + 0.5Z_2 \tag{44}$$

*s.t.*

$$\sum_{i=1}^{m} x_{ij} \leq 1, j \in N \tag{45}$$

$$\sum_{j=1}^{n} x_{ij} = 1, i \in M \tag{46}$$

$$x_{ij} = 0 \, or \, 1, i \in M, j \in N \tag{47}$$

## 3. Results

### 3.1. Prediction of infections

Taking COVID-19 in Handan as an example, which occurred in April 2022, we could predict the number of cases. This paper uses the GM (1,1) model and the Metabolic GM (1,1) model to predict the number of infected people.

1). Data preprocessing

After summarizing the number of infected people in this epidemic, we can obtain the original sequence.

$$Q^{(0)} = (83, 81, 69, 67, 54, 31, 20)$$

Firstly, the model can select the first five data of the original sequence.

$$Q^{(0)} = (83, 81, 69, 67, 54)$$

According to Eq (3), its sequence of 1-AGO is as follows.

$$Q^{(1)} = (83, 164, 233, 300, 354)$$

From formula (5), its adjacent mean generating sequence is as follows.

$$Z^{(1)} = (123.5, 198.5, 266.5, 327)$$

We can get results.

$$B = \begin{bmatrix} -123.5 & 1 \\ -198.5 & 1 \\ -266.5 & 1 \\ -327 & 1 \end{bmatrix}, Y = \begin{bmatrix} 81 \\ 69 \\ 67 \\ 54 \end{bmatrix},$$

Substitute into the formula (6), we get the equation.

$$\bar{a} = (B^T B)^{-1} B^T Y = \begin{bmatrix} a \\ b \end{bmatrix} = \begin{bmatrix} 0.1219 \\ 95.6430 \end{bmatrix}$$

From Eq (11), the time response formula of GM (1,1) is as follows.

$$\begin{cases} Q^{(1)}(t+1) = -714e^{-0.12t} + 797 \\ Q^{(0)}(t+1) = Q^{(1)}(t+1) - Q^{(1)}(t) \end{cases}$$

(2) Average relative error of simulated values

When meeting this condition $t = 1, 2, 3, 4$, the simulated value sequence is as follows.

$$(\hat{q}^{(0)}(2), \hat{q}^{(0)}(3), \hat{q}^{(0)}(4), \hat{q}^{(0)}(5)) = (80.7388, 71.6089, 63.5114, 56.3296)$$

We obtained the result by substituting this into Eq (16) to obtain the sequence of simulated residual errors.

$$(\varepsilon^{(0)}(2), \varepsilon^{(0)}(3), \varepsilon^{(0)}(4), \varepsilon^{(0)}(5))$$
$$= (q^{(0)}(2), q^{(0)}(3), q^{(0)}(4), q^{(0)}(5)) - (\hat{q}^{(0)}(2), \hat{q}^{(0)}(3), \hat{q}^{(0)}(4), \hat{q}^{(0)}(5))$$

$$= (0.2612, -2.6089, 3.4886, -2.3296)$$

Substituting this into Eq (17), the average absolute percentage error (MAPE) is as follows.

$$\text{MAPE} = 2.6330\%$$

Because the model's MAPE value is negligible and meets the condition, the model's accuracy is good.

(3) Predict the number of infected people

When we meet the condition $k = 5$, we get the predicted value 49.9598.

$$\hat{q}^{(0)}(6) = 49.9598$$

The prediction residual is as follows.

$$g(6) = q^{(0)}(6) - \hat{q}^{(0)}(6) = -18.9598$$

MAPE is as follows.

$$\Delta_6 = 0.6116$$

By removing the oldest information $Q^{(0)}1$, we can insert new information $Q^{(0)}6$ and obtain the modelling sequence as follows.

$$Q^{(0)} = (81, 69, 67, 54, 31)$$

We can take the sequence $Q^{(0)}$ as the new initial sequence. Then we used the Metabolic GM (1,1) model twice. The simulated values and average absolute percentage errors are in Table 4.

Using the Metabolism GM (1,1) model after three iterations, the results are shown in Table 5. The raw data are grouped into five groups, and we have three iterations. The MAPE value for the third iteration is 6.7050% < 10%. The model passed the test.

**Table 4. Results of metabolic GM (1,1) model.**

| Iteration times | Development coefficient (a) | Grey action (b) |
|---|---|---|
| First iteration | 0.1219 | 95.6430 |
| Second iteration | 0.2112 | 99.0435 |
| Third iteration | 0.3725 | 108.3756 |

**Table 5. Third iterative prediction and test table for the original sequence.**

| Serial number | Original value | Predicted value |
|---|---|---|
| 1 | 69 | 69.0000 |
| 2 | 67 | 69.0206 |
| 3 | 54 | 47.5554 |
| 4 | 31 | 32.7658 |
| 5 | 20 | 22.5757 |
| Back 1 | | 15.5547 |
| Back 2 | | 10.7172 |
| Back 3 | | 7.3842 |
| MAPE | 6.7050% | |

In addition to the Metabolism GM (1,1) model, the GM (1,1) model is also introduced for data comparison. The data results for the GM (1,1) model are shown in Table 6. The MAPE value of the GM (1,1) model is 15.2260% > 10%. Therefore, its accuracy is not as high as Metabolic GM (1,1).

Fig 1 depicts the raw statistical data of infected people after the outbreak of COVID-19. And it also depicts infected people that predicted by the original GM (1,1) model. As seen in Fig 1, the overall curve of infected people with COVID-19 showed a downward trend. The decline confirmed on the first and second days has been relatively slow. It had a slow decline in the first two days because there were more cumulative cases from the first day of COVID-19, which took some time to cure. From the third to the sixth day, because most cases are treated systematically, infected people decreased, and the rate of decline is faster than the former. Moreover, from the sixth day, the speed slowed again because most cases had been treated, and new cases gradually reduced. The epidemic was nearing its end.

Fig 2 depicts the statistics of the original infected people. It predicted numbers by the GM (1,1) model and the Metabolic GM (1,1) model. As seen in Fig 2, in terms of the linear trend, the Metabolic GM (1,1) model is closer to fit the original data. The traditional GM (1,1) model's fitting trend is not close. Therefore, the Metabolic GM (1,1) model is more accurate than the traditional one.

From the Fig 3, the error of the GM (1,1) model is always more significant than that of the Metabolic GM (1,1) model in the first three data. However, in the fourth and fifth data, the

**Table 6. Results comparison of traditional GM (1,1) model.**

| Serial number | Original value | Predicted value |
|---|---|---|
| 1 | 83 | 83.0000 |
| 2 | 81 | 85.9505 |
| 3 | 69 | 69.5529 |
| 4 | 67 | 56.2836 |
| 5 | 54 | 45.5459 |
| 6 | 31 | 36.8567 |
| 7 | 20 | 29.8252 |
| Back 1 | | 24.1352 |
| Back 2 | | 19.5307 |
| Back 3 | | 15.8046 |
| MAPE | 15.2260% | |
| Development coefficient (a) | 0.2117 | |
| Grey action (b) | 112.9380 | |

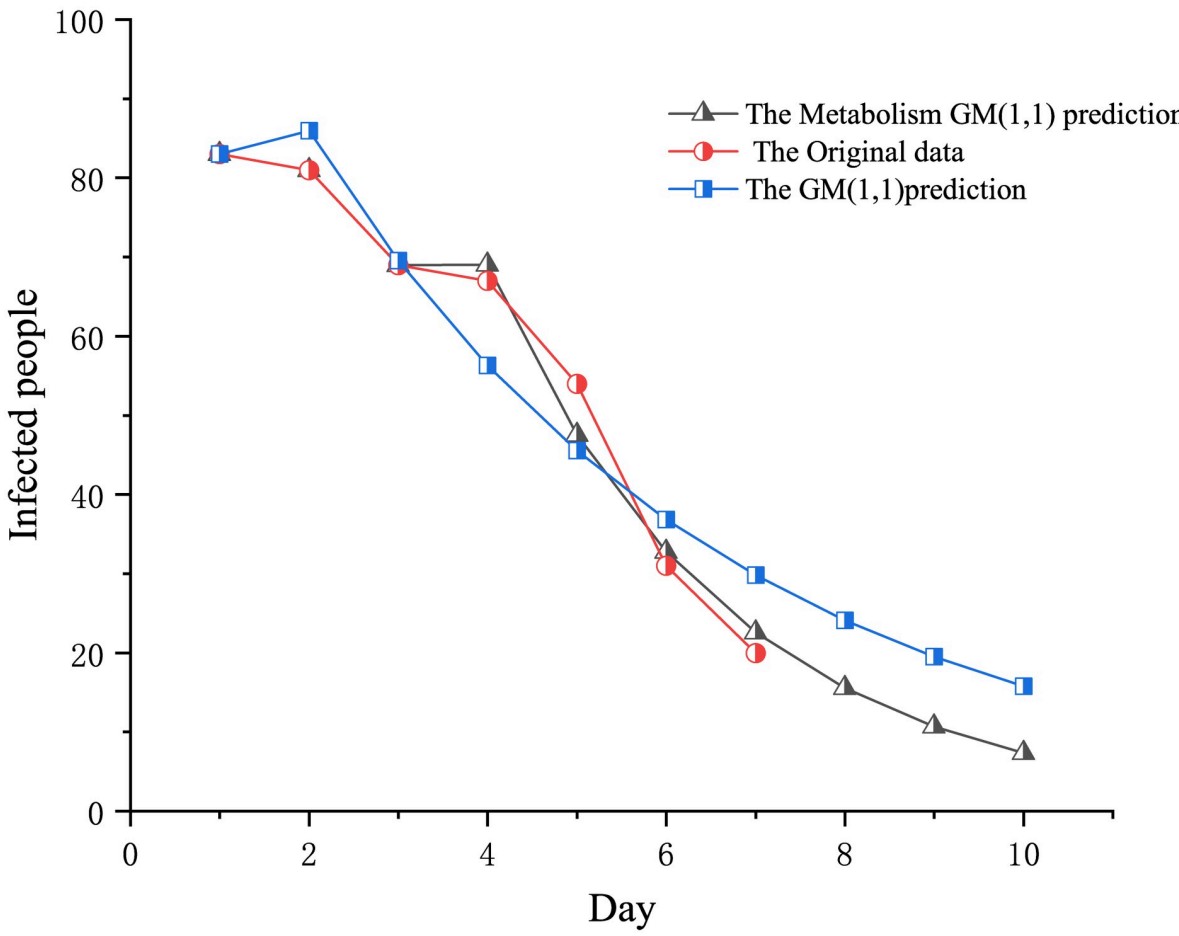

**Fig 1. Infected people of COVID-19.**

error of the Metabolic GM (1,1) model is slightly higher than that of the GM (1,1) model. After the fifth data, the difference between the two models becomes more significant, and the error of the Metabolic GM (1,1) model is smaller than that of the GM (1,1) model. So the Metabolic GM (1,1) model is more suitable.

Fig 4 depicts the forecast of infected people and the food demand in emergency supplies within 10 days. In Fig 4, when the growth time increased, the COVID-19 infection rate decreased, and the cumulative number of infected people increased, so the food supply also needs to be increased. Besides, the food demand required to calculate the highest demand is based on per capita food. The growth rate is the most significant on the first day and the second day, and the growth rate is slower between the second day and the tenth day. Moreover, the decline rate from the second to the seventh day was faster than from the seventh day to the tenth day. This phenomenon shows that with the implementation of drugs, vaccines, and epidemic prevention policies, COVID-19 has been effectively controlled, and the situation is developing positively.

### 3.2. Material demand results

We calculate each person's material demand according to the daily food consumption, which uses the material demand prediction formula considering the time difference. The predicted maximum, minimum, and likely demand results are shown in Table 7.

## Nonlinear fitting graph

**Fig 2. The nonlinear fitting graph.**

Table 7 describes the daily requirements for emergency supplies of food. Everyone's food consumption takes 300g as the minimum food demand and 500g as the highest daily demand. The difference between the minimum and maximum demand can be used as a reserve food amount to prevent phased outages. We could see that before the third day, we had controlled the epidemic, and infected people and food demand were rising. After the third day, we effectively controlled the epidemic, infected people, and material demand decreased.

Fig 5 depicts the evolution of infected people from the highest to the lowest. As seen in Fig 5, the maximum food demand and the minimum demand ($C$) are on the rise, but the rate of increase is getting slower. The number of infected people generally shows the opposite. In this study, the number of infected people has increased, and the material assurance is as high as possible. Therefore, the predicted number of infections is 16, 11, and 8, respectively, from the eighth to the tenth day.

### 3.3. Decision model result

1) The result of the entropy weight method

The supplier index matrix is:

$$R^* = \begin{bmatrix} 0.311 & 0.285 & 0.293 & 0.269 & 0.258 \\ 0.258 & 0.263 & 0.249 & 0.238 & 0.231 \\ 0.209 & 0.225 & 0.233 & 0.242 & 0.245 \\ 0.222 & 0.227 & 0.225 & 0.251 & 0.266 \end{bmatrix}$$

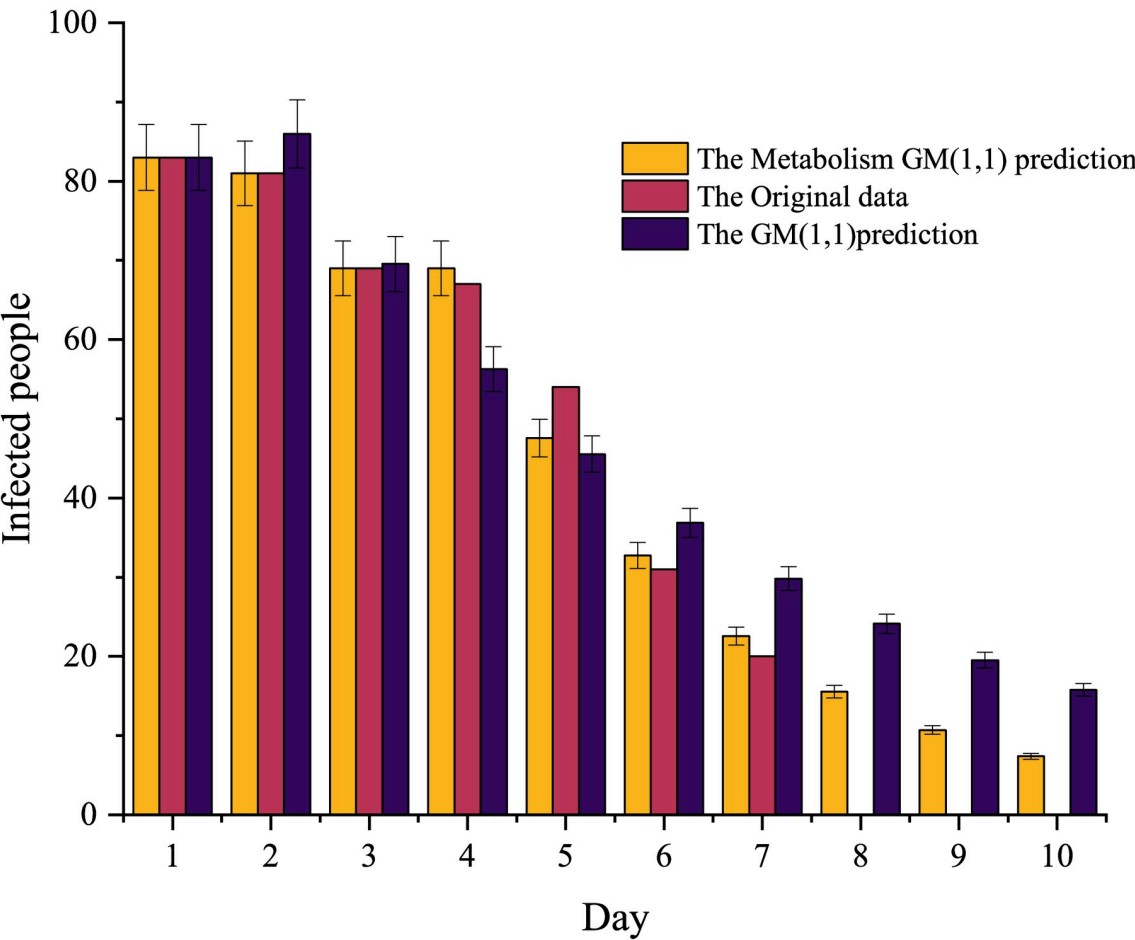

**Fig 3. Predicted infected people with error bars.**

The quantitative index matrix of the supplier is:

$$Y_{ij}^* = \begin{bmatrix} 1 & 0.7451 & 0.8235 & 0.5882 & 0.4804 \\ 0.4804 & 0.5294 & 0.3922 & 0.2843 & 0.2157 \\ 0 & 0.1569 & 0.2353 & 0.3235 & 0.3529 \\ 0.1275 & 0.1765 & 0.1569 & 0.4118 & 0.5588 \end{bmatrix}$$

The entropy values of each evaluation index are as follows.
First line:

$$p_{ij}^* = [0.2749, 0.2049, 0.2264, 0.1617, 0.1321]$$

Second line:

$$p_{ij}^* = [0.2526, 0.2783, 0.2062, 0.1495, 0.1134]$$

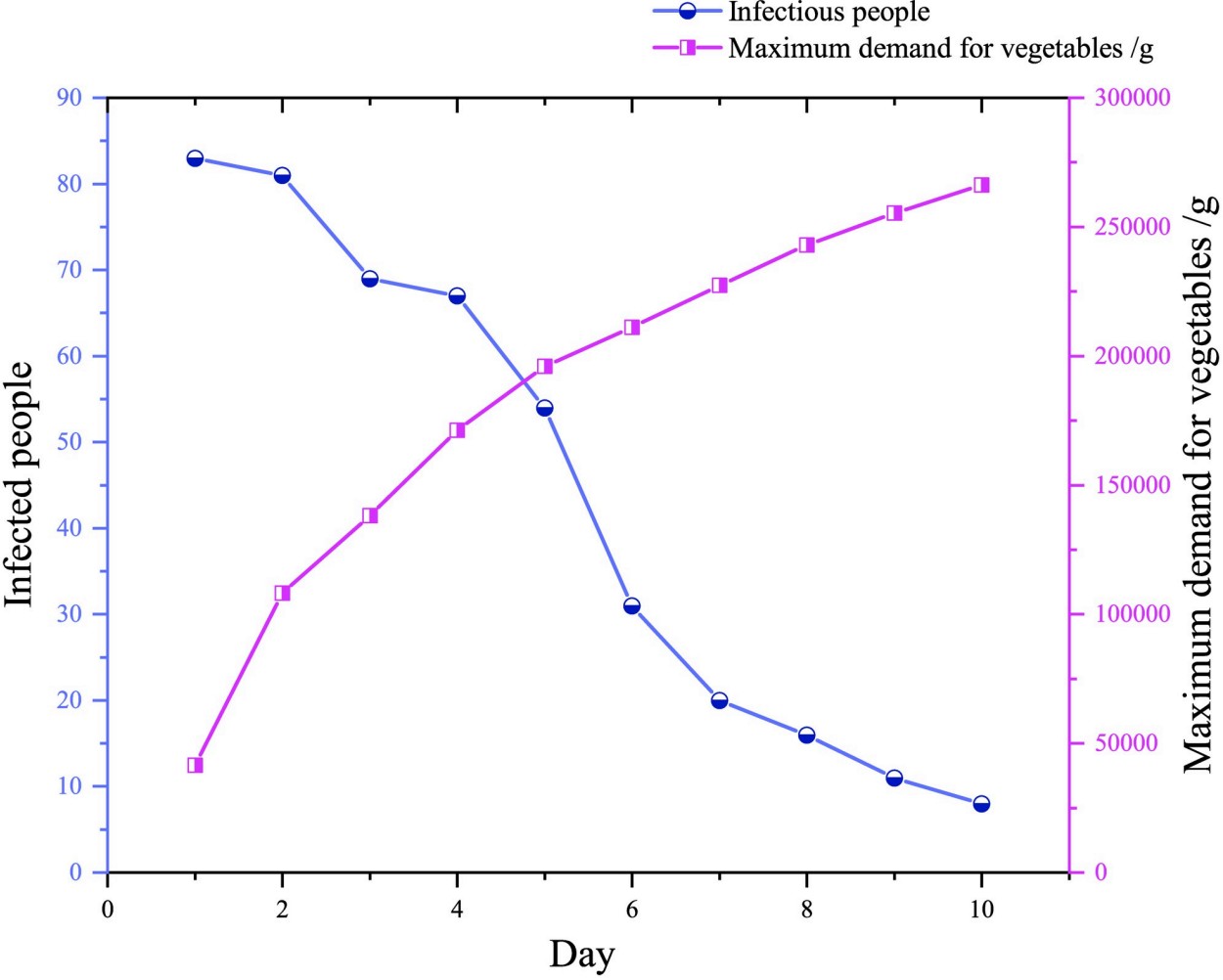

**Fig 4. Prediction of emergency supplies.**

**Table 7. Estimates of material demand.**

| Time | The number of infected | Food/g | | |
|------|------------------------|--------|--------|--------|
| | | **Minimum material demand** | **Maximum demand for materials** | **Potential material shortage** |
| 2 | 81 | 64893 | 108160 | 41000 |
| 3 | 69 | 82931 | 138219 | 38833.34 |
| 4 | 67 | 102720 | 171200 | 37500 |
| 5 | 54 | 117609 | 196026 | 35400 |
| 6 | 31 | 126730 | 211210 | 32083.34 |
| 7 | 20 | 136410 | 227350 | 28928.57 |
| 8 | 15.5547 | 145832.1 | 243059 | 26312.5 |
| 9 | 10.7172 | 153200.85 | 255343 | 24000 |
| 10 | 7.38420 | 159745.65 | 266234.5 | 22000 |

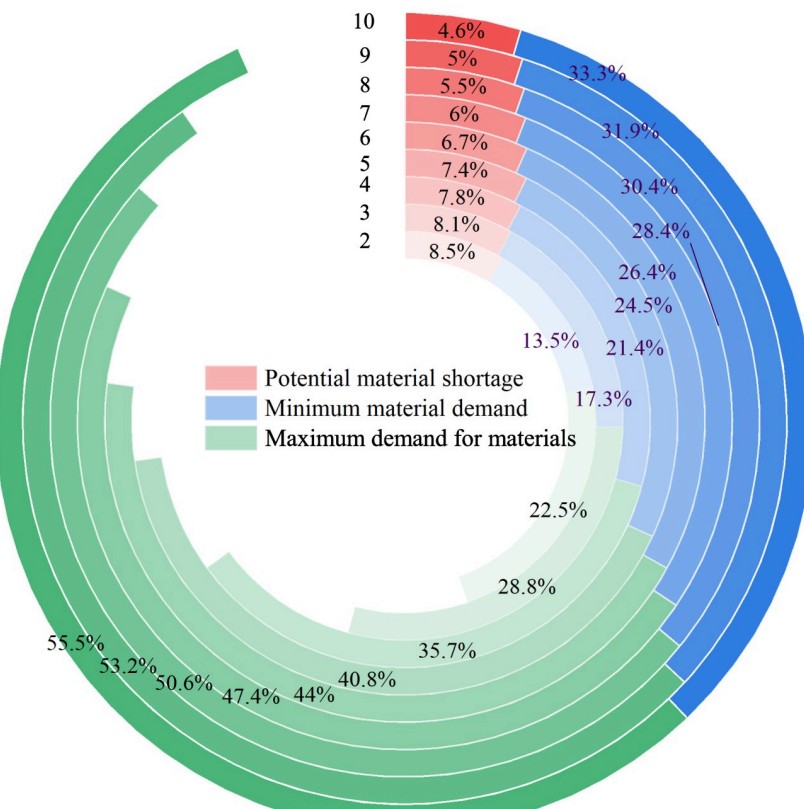

**Fig 5. Material demand estimates.**

Third line:

$$p_{ij}^* = [0, 0.1468, 0.2202, 0.3027, 0.3302]$$

Fourth line:

$$p_{ij}^* = [0.0891, 0.1233, 0.1096, 0.2877, 0.3904]$$

Index information entropy:

$$E_j^* = [-0.0425, -0.0438, -0.0541, -0.1108]$$

The weight of each index of the supplier:

$$W_i^* = [0.2452, 0.2455, 0.2480, 0.2613]$$

The original matrix of each index on the demand side:

$$R = \begin{bmatrix} 0.09 & 0.083 & 0.089 & 0.102 & 0.105 \\ 0.11 & 0.15 & 0.092 & 0.047 & 0.108 \\ 0.065 & 0.058 & 0.083 & 0.075 & 0.092 \\ 0.042 & 0.039 & 0.035 & 0.016 & 0.028 \\ 0.085 & 0.091 & 0.105 & 0.095 & 0.103 \\ 0.13 & 0.15 & 0.073 & 0.18 & 0.105 \\ 0.143 & 0.131 & 0.135 & 0.14 & 0.098 \\ 0.13 & 0.11 & 0.153 & 0.112 & 0.132 \\ 0.15 & 0.145 & 0.124 & 0.115 & 0.104 \\ 0.055 & 0.043 & 0.111 & 0.118 & 0.134 \end{bmatrix}$$

The demand side quantifies the index matrix:

$$Y_{ij} = \begin{bmatrix} 0.4512 & 0.4085 & 0.4451 & 0.5244 & 0.5427 \\ 0.5732 & 0.8171 & 0.4634 & 0.1890 & 0.5610 \\ 0.2988 & 0.2561 & 0.4085 & 0.3598 & 0.4634 \\ 0.1585 & 0.1402 & 0.1159 & 0 & 0.0732 \\ 0.4207 & 0.4573 & 0.5427 & 0.4817 & 0.5305 \\ 0.6951 & 0.8171 & 0.3476 & 1 & 0.5427 \\ 0.7744 & 0.7012 & 0.7256 & 0.7561 & 0.5000 \\ 0.6951 & 0.5732 & 0.8354 & 0.5854 & 0.7073 \\ 0.8171 & 0.7866 & 0.6585 & 0.6037 & 0.5366 \\ 0.2378 & 0.1646 & 0.5793 & 0.6220 & 0.7195 \end{bmatrix}$$

Each index entropy of the demand side is as follows.
First line:

$$p_{ij} = [0.1902, 0.1722, 0.1877, 0.2211, 0.2288]$$

Second line:

$$p_{ij} = [0.2201, 0.3138, 0.1780, 0.0726, 0.2155]$$

Third line:

$$p_{ij} = [0.1672, 0.1433, 0.2286, 0.2014, 0.2594]$$

Fourth line:

$$p_{ij} = [0.3249, 0.2874, 0.2376, 0, 0.1501]$$

Fifth line:

$$p_{ij} = [0.1729, 0.1880, 0.2231, 0.1980, 0.2181]$$

**Table 8. The supplier's evaluation of material requirements urgency.**

| $L_n$ | $C_1$ | $C_2$ | $C_3$ | $C_4$ | $C_5$ | $C_6$ | $C_7$ | $C_8$ | $C_9$ | $C_{10}$ |
|------|------|------|------|------|------|------|------|------|------|------|
| $L_1$ | 3 | 4 | 1 | 2 | 2 | 2 | 3 | 2 | 2 | 4 |
| $L_2$ | 5 | 1 | 3 | 2 | 1 | 1 | 3 | 3 | 2 | 4 |
| $L_3$ | 2 | 2 | 2 | 4 | 3 | 3 | 4 | 2 | 3 | 3 |
| $L_4$ | 4 | 3 | 2 | 5 | 5 | 2 | 2 | 4 | 3 | 3 |
| $L_5$ | 1 | 3 | 4 | 5 | 4 | 1 | 3 | 1 | 4 | 2 |

Sixth line:

$$p_{ij} = [0.2043, 0.2401, 0.1022, 0.2939, 0.1595]$$

Seventh line:

$$p_{ij} = [0.2240, 0.2028, 0.2099, 0.2187, 0.1446]$$

Eighth line:

$$p_{ij} = [0.2047, 0.1688, 0.2460, 0.1724, 0.2082]$$

Ninth line:

$$p_{ij} = [0.2401, 0.2312, 0.1935, 0.1774, 0.1577]$$

Tenth line:

$$p_{ij} = [0.1024, 0.0709, 0.2494, 0.2677, 0.3097]$$

Index information entropy:

$$E_j = [-0.0405, -0.0461, -0.0417, -0.0534, -0.0404, -0.0443, -0.0408, -0.0408, -0.0410, -0.0491]$$

The weight of each indicator on the demand side:

$$W_i = [0.0997, 0.1002, 0.0998, 0.1009, 0.0997, 0.1000, 0.0997, 0.0997, 0.0997, 0.1005]$$

2) Profit and loss matrix calculations results

The evaluation results of emergency materials from both supplier and demander are shown in Table 8 and Table 9.

As shown in Table 8, $C_l$ is the emergency supplies' attributes. $L_n$ is five suppliers. Table 8 represents the five suppliers' evaluation of the attributes of the emergency materials.

As shown in Table 9, $C_l$ is the emergency demands' attributes. $T_n$ is five demanders. Table 9 represents the five demanders' evaluation of the attributes of the emergency materials.

**Table 9. The demander's expectation of the urgency of the material requirements.**

| $T_n$ | $C_1$ | $C_2$ | $C_3$ | $C_4$ | $C_5$ | $C_6$ | $C_7$ | $C_8$ | $C_9$ | $C_{10}$ |
|------|------|------|------|------|------|------|------|------|------|------|
| $T_1$ | 3 | 2 | 2 | 3 | 2 | 4 | 1 | 4 | 3 | 2 |
| $T_2$ | 2 | 1 | 3 | 3 | 2 | 4 | 3 | 5 | 2 | 2 |
| $T_3$ | 2 | 2 | 1 | 2 | 1 | 3 | 2 | 3 | 4 | 3 |
| $T_4$ | 2 | 3 | 2 | 2 | 1 | 4 | 2 | 1 | 4 | 4 |
| $T_5$ | 1 | 3 | 4 | 4 | 3 | 4 | 3 | 2 | 3 | 4 |

3) Perceived utility matrix calculation results

According to the evaluation of material demand urgency by both supplier and demander, from the formula (26) to (38), the comprehensive perceived utility matrix $V$ and $V'$ are obtained.

$$V = v_{ij}^b = \begin{bmatrix} 0.1015 & 0.1084 & 0.0627 & 0.0688 & 0.1135 \\ 0.1063 & 0.0713 & 0.0842 & 0.1145 & 0.1165 \\ 0.0723 & 0.0827 & 0.0676 & 0.0980 & 0.1075 \\ 0.0724 & 0.0939 & 0.0803 & 0.1055 & 0.0941 \\ 0.0987 & 0.1147 & 0.0487 & 0.0869 & 0.0540 \end{bmatrix}$$

$$V' = v_{ij}^s = \begin{bmatrix} 0.1488 & 0.1628 & 0.0441 & 0.0625 & 0.0505 \\ 0.1285 & 0.1100 & 0.0575 & 0.1027 & 0.0542 \\ 0.1284 & 0.1441 & 0.0496 & 0.0961 & 0.0585 \\ 0.0813 & 0.1337 & 0.0962 & 0.1061 & 0.0656 \\ 0.0961 & 0.1503 & 0.0607 & 0.0723 & 0.0593 \end{bmatrix}$$

4) The results of the bi-objective optimization model

In this section, the bi-objective optimization is transformed into single objective optimization model. And the objective function is solved by genetic algorithm. The algorithm parameters are as follows: population number is 10, crossover probability is 0.8, mutation probability is 0.2, iteration book is 100.

The maximum value of objective function is 0.1236, the maximum value of objective function one is 0.4315, and the maximum value of objective function two is 0.4941. The best matching strategy is that supplier $L_1$ supplies community $T_1$, supplier $L_2$ supplies community $T_4$, supplier $L_3$ supplies community $T_3$, supplier $L_4$ supplies community $T_2$, and supplier $L_5$ supplies community $T_5$. In this way, both sides can optimize satisfaction degree. They also can take the material demand urgency into account. The decision matrix is shown in $x$.

$$x = \begin{bmatrix} 1 & 0 & 0 & 0 & 0 \\ 0 & 0 & 0 & 1 & 0 \\ 0 & 0 & 1 & 0 & 0 \\ 0 & 1 & 0 & 0 & 0 \\ 0 & 0 & 0 & 0 & 1 \end{bmatrix}$$

In this research, we used the genetic algorithm to solve the final matching result. The population evolution curve of the genetic algorithm is shown in Fig 6. The objective function has reached the optimal value. The fitness function converges slowly. Moreover, Fig 6 is the minimum of the objective function, and the maximum of the objective function is 0.1236.

## 4. Discussion

This paper uses the entropy weight method to evaluate the index matrix of materials demand urgency from both the supplier and the demander. Each index weight of the supplier is

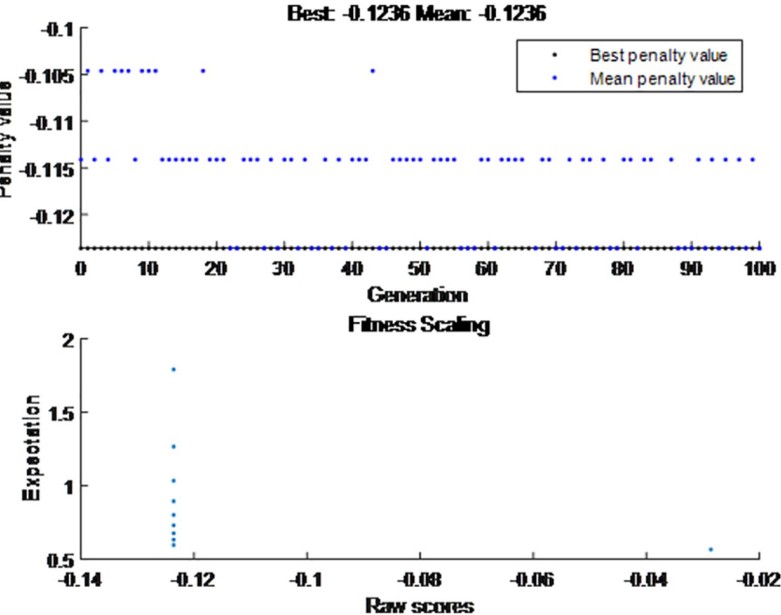

**Fig 6. Evolutionary curve of genetic algorithm.**

$W_i^* = [0.2452, 0.2455, 0.2480, 0.2613]$, and each index weight of the demander is

$$W_i = [0.0997, 0.1002, 0.0998, 0.1009, 0.0997, 0.1000, 0.0997, 0.0997, 0.0997, 0.1005].$$

From the supplier index weight, we can see that the order of each index in materials demand urgency is from large to small, such as distribution time, speed, supply capacity, and location. The delivery time is the most important to the supplier because the shorter the delivery time, the faster the speed. From the demand index weight, we can see that the ranking of each index of the material demand urgency is from large to small. Moreover, the factors are the number of people, distribution time, supply capacity, emergency materials reserve, number of difficult people, location, number of infected people, time of out-of-stock of irreplaceable goods, transportation convenience, and distribution cost. It can be seen from the total number of the material demand urgency because the total demand is the direct cause of the change in the material demand. The greater the number of people needed, the greater the urgency of material needs.

The weight of both sides uses 0.5 to uphold the principle of fairness and justice. However, the proportion of supply and demand is not necessarily the same in the matching decision. In order to further verify the model's validity, we change the satisfaction degree of the supplier and the demander from 0 to 1, respectively.

As shown in Fig 7, $Z_1$ is on the demand side, and $Z_2$ is on the supply side. The proportion of the demand side is inversely proportional to the satisfaction of the objective function, and the proportion of the supply side is directly proportional to the satisfaction of the objective function. The main reason may be the fewer attributes of the supplier indicators. However, the different weights of supply and demand do not affect the supply and demand matching results. The model can quickly get the best matching result under the premise of considering the urgency of emergency material demand and the satisfaction of both the supplier and the demander, which is beneficial to improve the fairness of distribution.

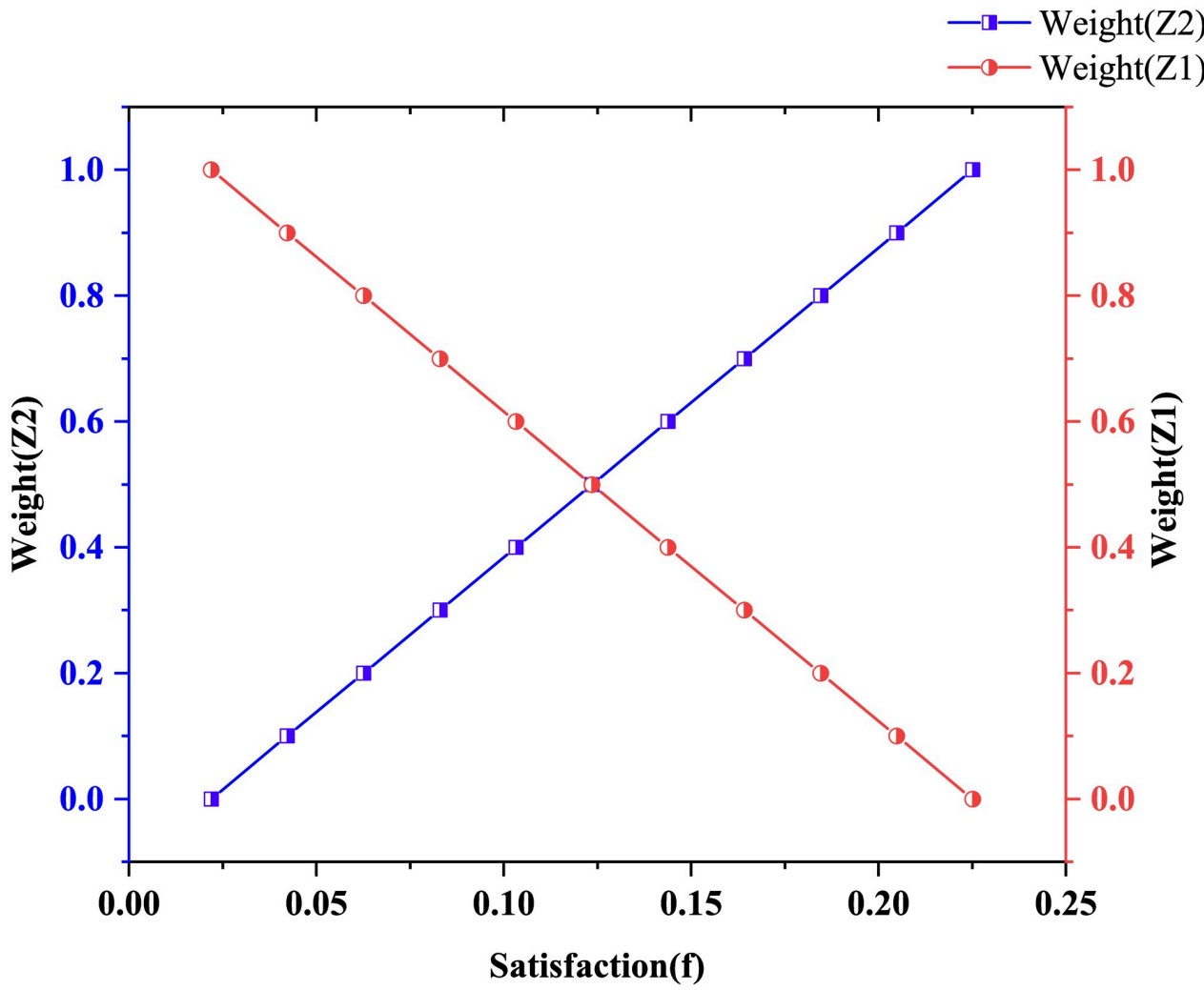

**Fig 7. The influence of supply and demand index weight on the total target.**

## 5. Conclusions

In this study, we used a metabolic GM (1,1) model to predict the number of infections in five communities and five distribution sites in the same region of Handan in the context of a public health emergency. At the same time, this research used the material demand forecast formula to forecast the emergency material demand based on the number of infected people. Besides, the research also solved the matching supply and demand problem based on the material demand urgency and supplier and demander satisfaction. This paper constructed the supplier and the demander's evaluation index system of material demand urgency. By constructing the profit and loss matrix and the perceived utility matrix, the optimal matching result was obtained by genetic algorithm.

The significance of this paper is to distinguish emergency materials' urgency, considering the satisfaction of both supply and demand. On the one hand, the matching result can reduce the cost of emergency logistics and improve the efficiency of deliver emergency materials. On the other hand, distinguishing emergency materials' urgency is beneficial to popularize the application in other emergencies, and improve the fairness of resource allocation.

Although this paper has achieved some research results, there are still some things that could be improved. In this study, the decision-making method of supply-demand matching only considers material demand urgency and the satisfaction of both the supplier and the demander. Other conditions can be added to perfect the relevant decision-making in the future. In addition, with supply and demand matching in the same region, the future can join the study of different regions to improve suppliers' and demanders' matching decision-making.

## Supporting information

**S1 File.**
(DOCX)

## Acknowledgments

The authors would like to thank the editors and the anonymous reviewers whose insightful comments have helped to improve the quality of this paper considerably.

## Author Contributions

**Conceptualization:** Zhichao Ma, Jie Zhang.

**Data curation:** Jie Zhang.

**Formal analysis:** Zhichao Ma, Jie Zhang.

**Funding acquisition:** Zhichao Ma.

**Investigation:** Jie Zhang.

**Methodology:** Jie Zhang.

**Software:** Jie Zhang.

**Validation:** Jie Zhang.

**Visualization:** Zhichao Ma, Jie Zhang.

**Writing – original draft:** Jie Zhang.

**Writing – review & editing:** Zhichao Ma, Jie Zhang, Shaochan Gao.

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
