## [Decision Letter · Decision Letter 0]

21 Dec 2022

PONE-D-22-31654Emergency supplies demand research under the background of COVID-19PLOS ONE

Dear Dr. Jie Zhang,

Thank you for submitting your manuscript to PLOS ONE. After careful consideration, we feel that it has merit but does not fully meet PLOS ONE’s publication criteria as it currently stands. Therefore, we invite you to submit a revised version of the manuscript that addresses the points raised during the review process.

A major revision is recommend incorporating a point-to-point authors reply.==============================

We look forward to receiving your revised manuscript.

Kind regards,

Mintode Nicodeme Atchade

Academic Editor

PLOS ONE

Journal Requirements:

3. We note you have included a table to which you do not refer in the text of your manuscript. Please ensure that you refer to Tables 6 and 7 in your text; if accepted, production will need this reference to link the reader to the Table.

Reviewers' comments:

Reviewer's Responses to Questions

**Comments to the Author**

1. Is the manuscript technically sound, and do the data support the conclusions?

Reviewer #1: Yes

Reviewer #2: Yes

2. Has the statistical analysis been performed appropriately and rigorously? 

Reviewer #1: Yes

Reviewer #2: Yes

3. Have the authors made all data underlying the findings in their manuscript fully available?

Reviewer #1: Yes

Reviewer #2: Yes

4. Is the manuscript presented in an intelligible fashion and written in standard English?

Reviewer #1: Yes

Reviewer #2: Yes

5. Review Comments to the Author

Reviewer #1: 1-I have read the paper very carefully but I found some points that should be done

The novelty should be written in a clear way.

2-The main contribution of the paper is not clear should be enhanced in a single paragraph.

3- The paper is interesting and can be enhanced and published after making the upcoming comments.

4-The idea of the paper is good and has merits.

5) Even though the authors have a very technically sound paper, it still lacking in depth of communication. Except its meant for select audience, the paper would benefit from simplification and rewording. As a statistician I am able to digest a lot of what is being done however, if your desire is to see the application of the work widely then more effort needs to come in to help audience understand this highly technically sound paper. Again authors also need to check for grammatical errors in the paper.

6- The "Abstract" of the paper. I consider that the authors should structure the "Abstract" as to cover the most important points of interest: the authors should have positioned the manuscript's topic in a broad context therefore covering appropriately the topic's background; the authors should have presented succinctly the methods they have employed in order to attain the purpose of their study;

7-The introduction is too long.

8-The abstract and conclusion are too long.

9-The numbers of equations is not correct.

I can accept the paper after making these comments

The changes made based on the comments should be written in color. After carrying out these changes, I recommend that the paper can be published.

Reviewer #2: There are a lot of typos that need to be corrected.

The introduction must be reorganized as it is very poor.

The novelty of the article should be justified.

Please define acronyms the first time they appear in the text.

The English writing needs to improve! I have corrected only two pages in the pdf attached file. Check the attached pdf file for the "English corrections," Please do it for the rest of the pages. I could not check the rest pages.

It has several flaws! You should try to present all the achievements in a clear presentation!!!

6. PLOS authors have the option to publish the peer review history of their article (what does this mean?). If published, this will include your full peer review and any attached files.

Reviewer #1: No

Reviewer #2: No

---

## [Author Response · Author response to Decision Letter 0]

26 Jan 2023

Response to the academic editor

We really appreciate your carefulness and conscientiousness. Your suggestions are really valuable and helpful for revising and improving our paper. According to your suggestions, we have made the following revisions to this manuscript.

(Editors comment 1): Please ensure that your manuscript meets PLOS ONE's style requirements, including those for file naming. The PLOS ONE style templates can be found at 

(Our response): Thank you very much for reviewing our manuscript. We have altered the manuscript in accordance with the requirements of the journal.

(Editors comment 2): We suggest you thoroughly copyedit your manuscript for language usage, spelling, and grammar. If you do not know anyone who can help you do this, you may wish to consider employing a professional scientific editing service.

(Our response): Thanks for your suggestion. We do invite a friend of ours to help polish our article. And we hope the revised manuscript could be acceptable to you. The details of my friend are as follows.

Name: Shaochan Gao1, #a

1 School of Management Engineering and business, Hebei University of Engineering, Handan, Hebei, China

#a Current Address: Hebei University of Engineering, Congtai District, Handan, Hebei, China

(Editors comment 3): We note you have included a table to which you do not refer in the text of your manuscript. Please ensure that you refer to Tables 6 and 7 in your text; if accepted, production will need this reference to link the reader to the Table.

(Our response): Thank you very much for reviewing our manuscript. We delete Table 6 and Table 7. Table 6 and Table 7 belong to the process. Their removal does not affect the overall structure and results of the article.

Response to Reviewer 1

According to the associate reviewers’ comments, we have made extensive modifications to our manuscript and supplemented extra data to make our results convincing. Thank you again for your positive comments and valuable suggestions to improve the quality of our manuscript. 

(Comment 1): I have read the paper very carefully but I found some points that should be done. The novelty should be written in a clear way.

(Our response): Thank you very much for your careful reading and recognition of our manuscript. We changed some points in the structure of the article and marked them in yellow. As in the seventh paragraph of the Introduction. 

We have revised the Innovation Point, and the Innovation Point into a separate paragraph. The details are in paragraph 6 of the Introduction. We marked it in green.

(Comment 2): The main contribution of the paper is not clear and should be enhanced in a single paragraph.

(Our response): Thank you very much for your careful reading. We revised the main contribution of the article. The specific content in section 5, Conclusion, paragraph 1. The details are highlighted in purple.

(Comment 3): The paper is interesting and can be enhanced and published after making the upcoming comments.

(Our response): Thank you very much for your careful reading and recognition of our manuscript. Your valuable comments have a great effect on improving the quality of our manuscript.

(Comment 4): The idea of the paper is good and has merits.

(Our response): Thank you very much for your careful reading and recognition of our manuscript.

(Comment 5): Even though the authors have a very technically sound paper, it still lacking in depth of communication. Except its meant for select audience, the paper would benefit from simplification and rewording. As a statistician I am able to digest a lot of what is being done however, if your desire is to see the application of the work widely then more effort needs to come in to help audience understand this highly technically sound paper. Again authors also need to check for grammatical errors in the paper.

(Our response): Thank you very much for your careful reading and recognition of our manuscript. In order to make the article more in-depth, this article has made the following changes(Table 1). 

In order to expand the scope of the article, this article will be ‘In the context of Covid-19’ to ‘In the context of emergency public events’. 

In order to make the audience understand this paper better, we transform the research of the urgency of the demand for emergency materials into the matching problem of supply and demand based on the urgency and satisfaction of the demand for emergency materials.

We are sorry for the grammatical mistakes in the paper. We do invite a friend of ours to help polish our article. And we hope the revised manuscript could be acceptable to you.

 

Table 1. Article in-depth revision contrast

Before modification 

The Grey Correlation-TOPSIS Model:

(1) Establish an index system;

(2) Defuzzification of fuzzy indicators;

(3) Weights corresponding to indicators;

(4) Prioritize the urgency of your needs. (1 supply point and 5 demand points) 

After modification

Decision model result:

(1) Establish the supply and demand indicators system;

(2) The index weight is calculated by the entropy weight method;

(3) Build a profit and loss matrix;

(4) Construct the perceived utility matrix;

(5) Establish the double objective optimization model;

(6) On the basis of distinguishing the degree of urgency of demand, it maximizes the degree of satisfaction of both the supplier and the demander. Then, 5 supply points and 5 demand points are matched to get the optimal matching scheme.

(Comment 6): The "Abstract" of the paper. I consider that the authors should structure the "Abstract" as to cover the most important points of interest: the authors should have positioned the manuscript's topic in a broad context therefore covering appropriately the topic's background; the authors should have presented succinctly the methods they have employed in order to attain the purpose of their study;

(Our response): Thank you very much for your careful reading of our manuscript. For the abstract, we changed the covid-19 context to the broader context of public health emergencies. 

We have also optimized and abbreviated the methods section of the Abstract. The details are highlighted in blue.

(Comment 7): The introduction is too long.

(Our response): Thank you very much for your careful reading of our manuscript. We have revised the introduction and marked it in different colors.

(Comment 8): The abstract and conclusion are too long.

(Our response): Thank you very much for your careful reading of our manuscript. We have revised the abstract and conclusion to shorten the length of the manuscript while optimizing the content.

(Comment 9): The numbers of equations is not correct.

(Our response): Thank you very much for your careful reading of our manuscript. I apologize for our mistake. We have simplified the model and modified the formula number without changing the structure of the article. Change numbers are highlighted in yellow.

(Comment 10): I can accept the paper after making these comments.

The changes made based on the comments should be written in color. After carrying out these changes, I recommend that the paper can be published.

(Our response): Thank you very much for your careful reading and recognition of our manuscript. We have carefully revised each of your comments. We write the parts that have been changed in color. Each of your valuable comments has greatly improved the quality of our articles.

Response to Reviewer 2

We really appreciate your carefulness and conscientiousness. Your suggestions are really valuable and helpful for revising and improving our paper. According to your suggestions, we have made the following revisions to this manuscript.

(Comment 1): There are a lot of typos that need to be corrected.

(Our response): Thank you very much for your careful reading of our manuscript. I'm sorry for our spelling mistake. We have invited our friends to help us correct the spelling mistakes in the manuscript.

(Comment 2): The introduction must be reorganized as it is very poor.

(Our response): Thank you very much for your careful reading of our manuscript. We have rewritten the Introduction and color-coded it. I hope that the revised introduction can gain your approval.

(Comment 3): The novelty of the article should be justified.

(Our response): Thank you very much for your careful reading of our manuscript. We have rewritten the innovation points of the article and highlighted them in green. The details are in paragraph 6 of the Introduction.

(Comment 4): Please define acronyms the first time they appear in the text.

(Our response): Thank you very much for your careful reading of our manuscript. We have defined the acronyms that appear for the first time in our manuscript and highlighted them in red.

(Comment 5): The English writing needs to improve! I have corrected only two pages in the pdf attached file. Check the attached pdf file for the "English corrections," Please do it for the rest of the pages. I could not check the rest pages.

(Our response): Thank you very much for your careful reading of our manuscript. I'm sorry for our English writing ability. We checked the English attachments in time and will ask our friends to Polish our articles. I hope the revised article can get your approval.

(Comment 6): It has several flaws! You should try to present all the achievements in a clear presentation!!!

(Our response): Thank you very much for your careful reading. We revised the main contribution of the article. The specific content in section 5, Conclusion, paragraph 1. The details are highlighted in purple.

---

## [Decision Letter · Decision Letter 1]

23 Feb 2023

Research on emergency material demand based on urgency and satisfaction under public health emergencies

PONE-D-22-31654R1

Dear Dr. Zhang,

We’re pleased to inform you that your manuscript has been judged scientifically suitable for publication and will be formally accepted for publication once it meets all outstanding technical requirements.

Kind regards,

Lu Peng

Academic Editor

PLOS ONE

Additional Editor Comments (optional):

Reviewers' comments:

Reviewer's Responses to Questions

**Comments to the Author**

1. If the authors have adequately addressed your comments raised in a previous round of review and you feel that this manuscript is now acceptable for publication, you may indicate that here to bypass the “Comments to the Author” section, enter your conflict of interest statement in the “Confidential to Editor” section, and submit your "Accept" recommendation.

Reviewer #1: All comments have been addressed

Reviewer #2: All comments have been addressed

2. Is the manuscript technically sound, and do the data support the conclusions?

Reviewer #1: Yes

Reviewer #2: Yes

3. Has the statistical analysis been performed appropriately and rigorously? 

Reviewer #1: Yes

Reviewer #2: Yes

4. Have the authors made all data underlying the findings in their manuscript fully available?

Reviewer #1: Yes

Reviewer #2: Yes

5. Is the manuscript presented in an intelligible fashion and written in standard English?

Reviewer #1: Yes

Reviewer #2: Yes

6. Review Comments to the Author

Reviewer #1: All comments have been done i suggest accepting the paper

All comments have been done i suggest accepting the paper

Reviewer #2: The authors made a The study of emergency logistics has also attracted

scholars’ attention. Therefore, matching emergency materials’ supply and demand

quickly, which meets urgency and satisfaction, is the purpose of this paper. This paper

used the Metabolism Grey Model (1,1) (GM (1,1)) and the material demand prediction

model to predict the number of infections and material demand. Besides, we

established a double objective optimization model by constructing a profit and loss

matrix and a comprehensive utility perception matrix. The results show that the method

is helpful in matching the optimal supply and demand decision quickly on the basis of

meeting urgency and satisfaction. The method is helpful in improving the fairness of

emergency material distribution, which could better protect people's livelihoods.

I accept the paper

7. PLOS authors have the option to publish the peer review history of their article (what does this mean?). If published, this will include your full peer review and any attached files.

Reviewer #1: No

Reviewer #2: No

---

## [Editor Report · Acceptance letter]

14 Mar 2023

PONE-D-22-31654R1 

Research on emergency material demand based on urgency and satisfaction under public health emergencies 

Dear Dr. Zhang:

I'm pleased to inform you that your manuscript has been deemed suitable for publication in PLOS ONE. Congratulations! Your manuscript is now with our production department. 

Kind regards, 

on behalf of

Dr. Lu Peng 

Academic Editor

PLOS ONE